# Influence of Radiation-Induced Displacement Defect in 1.2 kV SiC Metal-Oxide-Semiconductor Field-Effect Transistors

**DOI:** 10.3390/mi13060901

**Published:** 2022-06-07

**Authors:** Gyeongyeop Lee, Jonghyeon Ha, Kihyun Kim, Hagyoul Bae, Chong-Eun Kim, Jungsik Kim

**Affiliations:** 1Department of Electrical Engineering, Gyeongsang National University (GNU), Jinju 52828, Korea; rudduq1660@gnu.ac.kr (G.L.); jh990502@gnu.ac.kr (J.H.); 2Division of Electronics Engineering and Future Semiconductor Convergence Technology Research Center, Jeonbuk National University, Jeonju 54896, Korea; kihyun.kim@jbnu.ac.kr (K.K.); hagyoul.bae@jbnu.ac.kr (H.B.); 3Department of Control and Instrumentation Engineering, Gyeongsang National University (GNU), Jinju 52828, Korea; cekim@gnu.ac.kr

**Keywords:** displacement defect, radiation effect, SiC MOSFET, TCAD simulation

## Abstract

The effect of displacement defect on SiC metal-oxide-semiconductor field-effect transistors (MOSFETs) due to radiation is investigated using technology computer-aided design (TCAD) simulation. The position, energy level, and concentration of the displacement defect are considered as variables. The transfer characteristics, breakdown voltage, and energy loss of a double-pulse switching test circuit are analyzed. Compared with the shallow defect energy level, the deepest defect energy level with *E_C_* − 1.55 eV exhibits considerable degradation. The on-current decreases by 54% and on-resistance increases by 293% due to the displacement defect generated at the parasitic junction field-effect transistor (JFET) region next to the P-well. Due to the existence of a defect in the drift region, the breakdown voltage increased up to 21 V. In the double-pulse switching test, the impact of displacement defect on the power loss of SiC MOSFETs is negligible.

## 1. Introduction

Wide-bandgap silicon carbide (SiC) is a promising material for power applications due to high breakdown strength, larger thermal conductivity, and high electron saturation velocity [1,2,3,4]. Therefore, silicon carbide metal-oxide-semiconductor field-effect transistors (SiC MOSFETs) can reduce volume and energy loss of a high power system [5]. Accordingly, SiC MOSFETs have been used in various applications, including automotives, aerospace, electric vehicle charging infrastructure, power supply, and unmanned aerial vehicles. However, previous studies [6,7,8,9,10,11] have revealed problems due to radiation in SiC MOSFETs. Problems such as an increased leakage current due to heavy ions [6] or threshold voltage shift due to the total ionizing dose (TID) [7], as well as catastrophic damage such as single-event burnout (SEB) [8,9,10,11] have been observed. Therefore, reliability in radiation environments is important for the stable operation of various applications.

SiC MOSFETs are continuously exposed to cosmic radiation not only in space, but also on the Earth’s surface [12]. High energy neutrons greater than 100 MeV as well as thermal neutrons are often found outdoors [12]. The average value of threshold displacement energy is 41 eV for silicon and 16 eV for carbon [13]. Therefore, the displacement damage in an SiC crystal would be accumulated overtime in terrestrial applications such as electrical vehicles, industry motor drives, and solar inverters. Furthermore, in particle-enriched environments, SiC MOSFETs become even more vulnerable to radiation, leading to device degradation and system failure [6,7,8,9,10,11]. 

The most typical effects produced by radiation can be divided into two categories: soft errors causing temporary failures and hard errors leading to permanent failures. The displacement defects are cumulative and permanent; thus, the devices could gradually degrade. When neutron particles penetrate into SiC material and collide with lattice atoms, the momentum transfer knocks off interstitial atoms Si or C from their original site [14]. Such displacement plays a role as a defect or defect states. If the displacement defects are formed near the active region relevant to the device operation, the consequence is performance degradation [15,16,17], such as on-current drop, on-resistance increase, and breakdown voltage reduction. Studies have shown that this degradation is also seen in high-energy heavy ion implantation of SiC devices [18]. Unfortunately, there is no efficient method to shield the device from a neutron strike. Although SiC devices have attracted much attention in autonomous driving vehicles, their long-term reliability due to displacement damage must be a very important topic for safety issues. If a displacement defect occurs in SiC devices used for motor drivers, the motor of an autonomous vehicle may cause abnormal operation. This malfunction is a big problem directly related to life. However, the immunity of SiC MOSFETs to the terrestrial neutrons remains largely uncharted territory. Research on neutrons in SiC MOSFETs is limited to single-event burnout (SEB) [8,9,10,11], and there is currently little information on displacement defects.

In this study, the effect of displacement defects on SiC power MOSFETs is investigated by technology computer-aided design (TCAD) simulation [19]. Defects can be randomly generated by radiation with various concentrations and locations. In actual irradiation experiments, it is very difficult to create a constant defect concentration in a small specific area. In the previous simulation-based literature [20], it is reported that most of the deterioration due to displacement defects in power devices occurs where current flows. Therefore, we set the displacement defects only in the region where the current flows. In this article, we investigate vulnerability and tolerance to specific locations and provide physical insight into displacement defects.

## 2. Simulation Modeling Methodology

This study adopted the Shockley–Read–Hall (SRH) and Auger recombination models, as well as the Canali model for high-field saturation and the Okuto model for impact ionization. For SiC material properties, anisotropic mobility and incomplete ionization were considered [21]. The displacement defect was modeled with six different energy levels in SiC materials: 0.69, 0.72, 0.88, 1.03, 1.08, and 1.55 eV below conduction band edge (*E_C_*) [16]. The device structure used in this simulation study referred to the structure of Wolfspeed (Cree)’s 1.2-kV SiC power MOSFET (C2M0080120D), illustrated in Figure 1a [22]. The device dimensions were calibrated with scanning electron microscope (SEM) imaging [23]. Figure 1b,c show *I_d_*–*V_g_* and *I_d_*–*V_d_* characteristics calibrated from the experimental data from [24]. The test device was a vertical N-Channel power MOSFETs with a planar gate structure. The channel was located in the P-well region, between the N+ region and top of the junction field-effect transistor (JFET) region. The overall device size was 20 µm on the X-axis (vertical) and 10 µm on the Y-axis (horizontal). The lengths of the P-well region, N+ region, and N-drift region were 0.5 µm, 0.2 µm, and 10 µm, respectively. Additionally, the channel length was 0.96 µm and the length of the JFET region was 2.4 µm.

## 3. Result and Discussion

### 3.1. DC Characteristics

Figure 2a shows various defect positions considered with a defect concentration (*N_d_*) of /10^19^ cm^3^: source region A, channel region B, JFET region C, drift region D, and drift region E. Clustered defects are defined as rectangular in shape with a width of 1.0 μm and height of 0.6 μm. *I_d_*–*V_g_* characteristics at *V_DS_* = 20 V with various defect positions and with a defect energy level of *E_C_* − 1.55 eV are shown Figure 2b. Among the different positions, position C located near the gate oxide in the JFET region resulted in the largest reduction of the on-state current. The on-state current drop for defect position C was about 54%. Position B with counter-doping (p-type) showed a slightly smaller reduction (~50%). As the positions of B and C were directly under the gate oxide, they easily perturbed the formation of the accumulation layer by the gate bias. In positions D and E, the slope of the on-current started to decrease and the drain current was saturated for *V_g_* > 10 V and *V_g_* > 12 V, respectively. As the location was relatively far from the gate oxide interface, the charge trapping became effective at a higher gate voltage. Figure 2c shows *I_d_*–*V_d_* output characteristics at *V_GS_* = 20 V. The results indicate that the on-resistances were also degraded according to the defect position. Similar to the on-state current degradation, position C showed the greatest on-resistance degradation. At position B, where a 10% on-current reduction occurred, the defect was sufficiently controlled by the high gate bias (20 V).

Figure 3 shows the on-current and on-resistance dependence on different defect positions and their energy levels. The on-current was extracted from the *I_d_–V_g_* transfer characteristics at *V_g_* = 14 V and on-resistance was extracted at *V_d_* flowing Id = 20 A at *V_g_* = 20 V. The previous literature [25] reported, using a deep-level transient spectroscopy (DLTS) analysis, that the defect concentration (*N_d_*) increases with increasing neutron fluences. Therefore, it is assumed *N_d_* = 10^19^/cm^3^ is caused by more neutron fluences than *N_d_* = 10^17^/cm^3^. The electron density contours are shown in Figure 4a for no defect and Figure 4b–f for five different defect locations. As shown in Figure 3a, the on-current was smaller as the defect concentration was larger. As the defects are similar to a Coulomb scattering center, the carrier mobility and on-current degradation degrade. The source region (position A) had the least degradation because heavy N^+^ forms inherently have a large electron density. Therefore, the electron density stays high with defects, as shown in Figure 4b. In the p-channel region (position B), the inverted electron density right under the oxide could be retained as relatively large for *N_d_* = 10^17^/cm^3^. However, as shown in Figure 4c, *N_d_* = 10^19^/cm^3^ can hamper the formation of the inversion channel and laterally transport through the narrow channel. The degradation was most significant when the defects were formed in the JFET region (positions C and D). Interestingly, the worst defect position varied with the defect concentration. The worst conditions were position D for *N_d_* = 10^17^/cm^3^ and position C for *N_d_* = 10^19^/cm^3^. Referring to Figure 4d,e, the electrons were extended from the narrow lateral channel to the broad vertical path. In other words, the large electron density spread as the carriers entered into the JFET region and the location near position C had a larger electron density than the location near position D. However, because the electron density near position C is inherently greater than 10^19^/cm^3^, the carrier depletion due to *N_d_* = 10^17^/cm^3^ in that region could be smaller compared to the impact of *N_d_* = 10^19^/cm^3^. As the defect concentration increased from 10^17^/cm^3^ to 10^19^/cm^3^, 17% and 45% reduction occurred at position D and position C, respectively. When the defect was formed far from the JEFT region (position E), the impact was reduced as the carrier bypassing path was formed on the left and right sides of the defects, as shown in Figure 4f.

The on-resistance with the different defect positions of both defect concentrations is shown in Figure 3b. The largest on-resistance shift showed a 293% increase for *N_d_* = 10^19^/cm^3^ at position C. Because the defect acts as a parasitic series resistor, the defect increases on-resistance. The foregoing was similar to that shown in Figure 3a, in which the on-current degradation was most severe at position D with *N_d_* = 10^17^/cm^3^, and at position C with *N_d_* = 10^19^/cm^3^.

As shown in Figure 3c, as the defect energy level deepened, the higher on-current degradation was found. Similarly, the on-resistance increased as the defects were positioned at deeper energy levels, as shown in Figure 3d. As a result, when the defect concentration was 10^19^/cm^3^ and the defect was located at position C, which was the deeper energy level (*E_C_* − 1.55 eV), the degradation was more severe. The impact of defect energy level on the performance can be understood through the energy band diagram laterally cut along the channel-to-JFET region underneath the gate oxide, as shown in Figure 5.

The energy diagram shows that the defect creates an energy barrier in the JFET region. The energy barrier makes it difficult for the free carrier to freely flow from the channel into the JFET region. Consequently, impeding the flow of electrons results in the deterioration of on-current and on-resistance. As the defect energy level becomes deeper, the barrier height becomes greater, the current decreases, and the on-resistance increases. The defect with the deep energy level is closer to the mid-band and occupies a low gate voltage, as shown in Figure 5. Hence, a defect with a deeper energy level causes a higher energy band and interrupts the electron movement.

In addition to the on-state current and on-resistance, breakdown voltage is an important figure of merit in power devices. The simulation was conducted to investigate the impact of defect position, energy level, and concentration on the breakdown voltage, as shown in Figure 6. The highest breakdown voltage was found at position E, as shown in Figure 6a. Since defects impede the flow of current, they increase the breakdown voltage. The defect energy level at which the breakdown voltage had the largest increase was *E_C_* − 1.55 eV; the larger the defect concentration, the greater the increase in breakdown voltage, as shown in Figure 6b. As a result, the deeper the defect energy level and the larger the defect concentration at position E, the higher the increase in breakdown voltage. However, the change from position A to E had a negligible effect on the change in defect concentration. Whereas the on-state current and on-resistance were severely degraded, the breakdown voltage tended to slightly increase with the defects. The increase in breakdown voltage was less than 2%. Thus, the breakdown voltage shift could be less of problem in practice. The reason is that impact ionization occurs due to a high drain voltage and high electric field, resulting in the generation of numerous electron-hole pairs. Therefore, localized region defects have little effect due to the numerous electron-hole pairs.

### 3.2. Double-Pulse Test

In order to investigate the impact of the defect on a switching circuit operation, a double-pulse test (DPT) circuit was configured by referencing [26] and the schematic is illustrated in Figure 7a. The simulated switching characteristics for a double-pulse input and the gate voltage, drain voltage, and drain current of SiC MOSFET DUT1 are illustrated in Figure 7b. Here, displacement defects were located in the JFET region (the worst condition). From the switching characteristics, the on-state (*E_ON_*) and off-state (*E_OFF_*) energy consumption values were calculated by integrating the product of the drain current and the voltage over the corresponding time intervals. Depending on the defect existence, the variations in the values of *E_ON_* and *E_OFF_* were 1.2 μJ and 0.5 μJ, respectively. The *E_ON_* decreased by 1.2 μJ because the defect depletes free electrons, as shown in Figure 4. However, the 1.2 μJ difference was negligible because it was under the most severe condition. The increase in *E_OFF_* was due to an increase in the leakage current caused by the defect. This was because the defect level of *E_C_* − 1.55 eV was close to the mid-bandgap and acted as a carrier generation center that increased the leakage current. However, the 0.6% increase in total energy loss due to the defects was insignificant.

In order to understand why the displacement defect is meaningless in the DPT, the electron density profiles without and with defect (the worst condition) in SiC MOSFETs are illustrated in Figure 8. The electron density profile was extracted at time = 1.7 × 10^−6^ μs. Although the electron density of the JFET region (near the gate oxide surface) was depleted by displacement defects in Figure 8b, the total drain current can be constant due to no change of electron profiles in the drift- and substrate-regions. It means that the displacement defect does not affect the switching characteristic of SiC MOSFETs and operates independently of power consumption.

## 4. Conclusions

In this study, we investigated the effect of radiation displacement defects on SiC MOSFETs using TCAD. The simulation analysis results indicate that the radiation displacement defects decreased the on-current and increased the on-resistance. The location mainly affected by the displacement defects is position C in the JFET region. In addition, the larger the defect concentration and the deeper the defect energy level, the more severe the performance degradation. The displacement defects decreased the on-current by 54% and increased the on-resistance by 293% under the worst displacement defect position. The significant decrease in carrier density due to neutron irradiation suggests that the displacement defects depleted the electrons in the SiC MOSFETs. The breakdown characteristics of SiC MOSFETs indicate that the displacement defects also contribute to the increase in breakdown voltage. However, this increase is only 1.4%; hence, the effect of displacement defect looks marginal. In the double-pulse test, the defect had negligible effects on switching loss.

## Figures and Tables

**Figure 1 micromachines-13-00901-f001:**
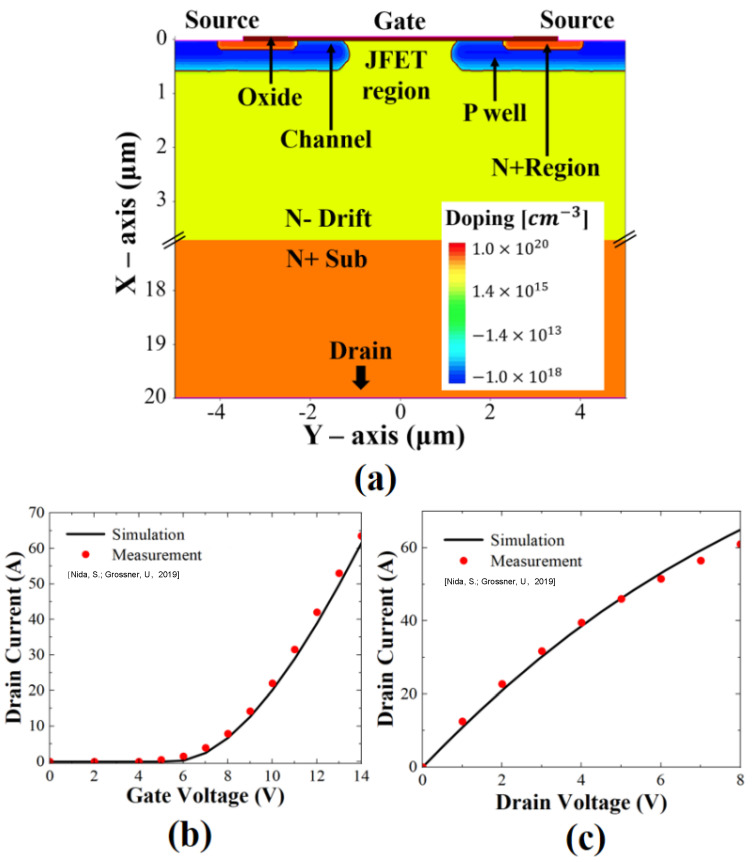
(**a**) Structure of simulated SiC MOSFET. Measured and simulated (**b**) *I_d_–V_g_* transfer and (**c**) *I_d_–V_d_* output characteristics for *V_DS_* = 20 V and *V_GS_* = 20 V, respectively.

**Figure 2 micromachines-13-00901-f002:**
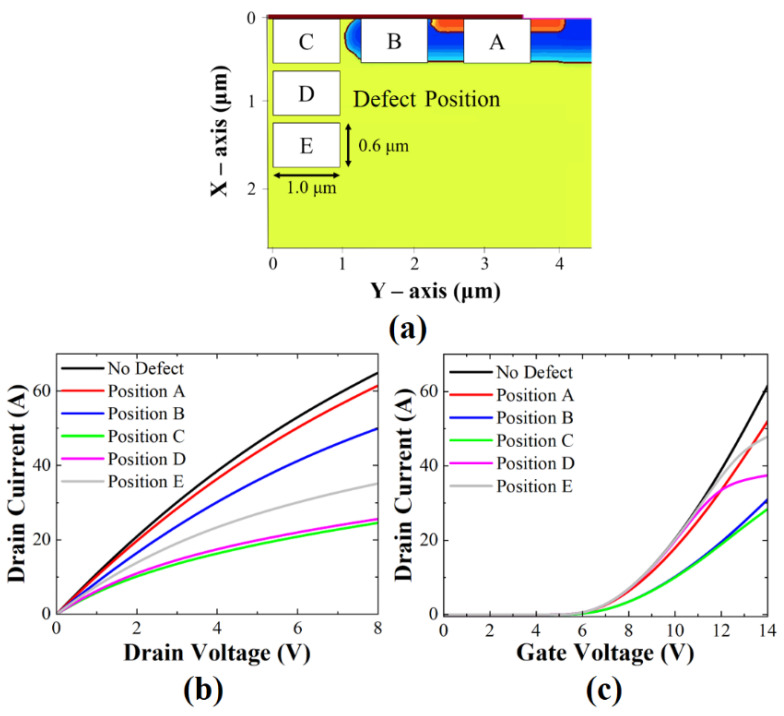
(**a**) Schematic of SiC MOSFET with displacement defects located at different positions. (**b**) Simulated *I_d_–V_g_* transfer characteristics at *V_DS_* = 20 V and (**c**) *I_d_–V_d_* output characteristics at *V_GS_* = 20 V with and without defect of various positions; defect energy level is fixed at *E_C_* − 1.55 eV, and defect concentration (*N_d_*) is 10^19^/cm^3^.

**Figure 3 micromachines-13-00901-f003:**
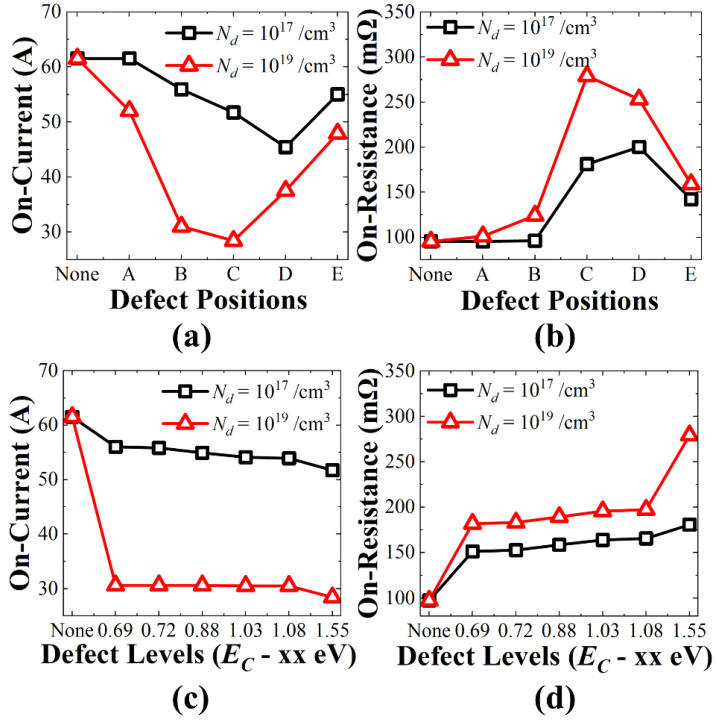
(**a**) On-current and (**b**) on-resistance variations with different defect positions; defect is acceptor-like defect with *E_C_* − 1.55 eV. (**c**) On-current and (**d**) on-resistance variations with various defect energy levels; defect location is position C.

**Figure 4 micromachines-13-00901-f004:**
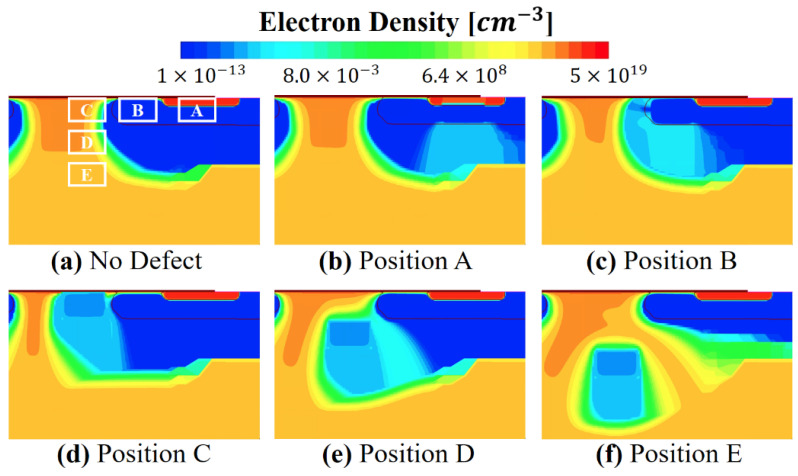
Electron density profiles of SiC MOSFETs (**a**) with no defect and (**b**–**f**) with accepter-like defects at five different locations. The bias condition is *V_GS_* = 14 V and *V_DS_* = 20 V. Defect energy level is *E_C_* − 1.55 eV.

**Figure 5 micromachines-13-00901-f005:**
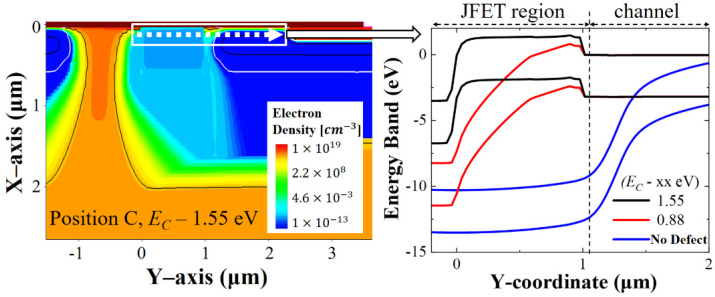
Electron density profile and energy band diagram from JFET to channel at gate oxide/silicon-carbide interface with *V_GS_* = 14 V and *V_DS_* = 20 V. Defects are located at position C (JFET region) with two energy levels; electron density profile and energy band diagram when defects do not exist are also indicated.

**Figure 6 micromachines-13-00901-f006:**
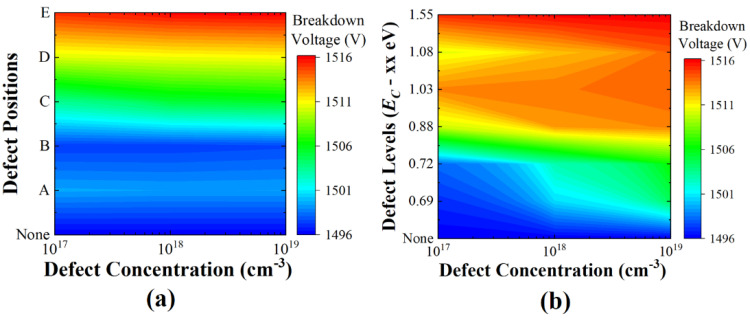
Effect of changes in (**a**) position and (**b**) concentration of displacement defects on breakdown voltage.

**Figure 7 micromachines-13-00901-f007:**
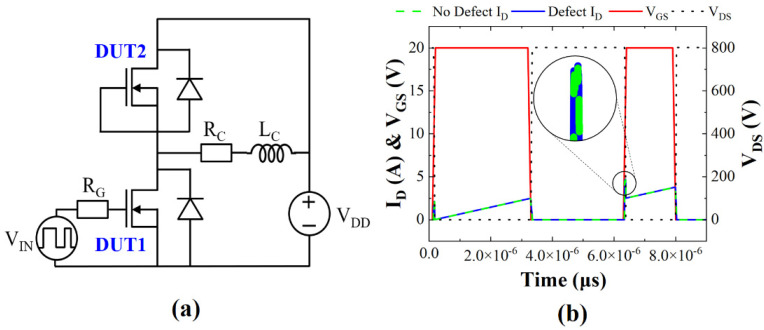
(**a**) Schematic of circuit used in double-pulse test. DUT1 and DUT2 are SiC MOSFET devices with antiparallel SiC p-n junction diode; *V_DD_* = 800 V and *V_IN_* (maximum voltage) = 20.0 V, *R_C_* = 1.0 Ω, *R_G_* = 10.0 Ω, and *L_C_* = 1000 μH. (**b**) Waveform of DUT1 SiC MOSFET with and without defect. The *E_C_* − 1.55 eV defect is located at position C (JFET region).

**Figure 8 micromachines-13-00901-f008:**
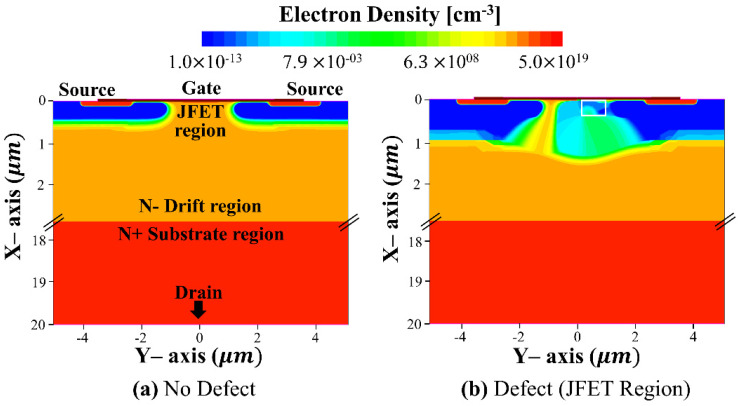
(**a**) Electron density profiles of SiC MOSFETs (**a**) with no defect and (**b**) with accepter-like defect (white line box). The bias condition is at *V_GS_* = 20 V and *V_DS_* = 0 V, when the time is 1.7 × 10^−6^ μs.

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
