# Peer review of "Influence of Radiation-Induced Displacement Defect in 1.2 kV SiC Metal-Oxide-Semiconductor Field-Effect Transistors"

_micromachines, 2022, doi:10.3390/mi13060901_

Round 1

Reviewer 1 Report

Authors have made efforts to simulate the impact of displacement defects on SiC based MOSFET using TCAD software. Impact of defect clusters at various positions in the device is simulated. The work may be interesting to community working in this area. However, there is lack of clarity in the work. Please see following comments and include them in revised version of the manuscript:

1. Previous Works related to Simulation based studies on the problem is missing and novel part is a vague. Please include.  SiC is a good material for harsh environmen temperature sensing applications: For example, foloowing work may be seen:   https://doi.org/10.1016/j.vacuum.2020.109590

2. Line 32 -- Include refrences after "Various studies". 

3. Line 3-- Rewrite it clearly. 

4. Line 41 -- include some examples of "terrestetial applications" . 

5 -- Line 52 -- "While----" Justify this statement. How autonomus vechiles affected and what are the safety issues? 

6. Include some most recent works on ion irradiation induced degradation in SiC material: For example, following work may be included:  https://doi.org/10.1016/j.matlet.2021.131150

7. Figure 1-- Please mention the dimensions of the device/regions for more clarity.

8. For figure 1 (b,c)-- Did authors took permission from Ref. 21? No credit line is included. 

9. Line 66- please include models for high field saturation. 

10. Line 74 -- Nchannel to N-Channel

11. Line 80 -- How the area of clustered trap is selected? Explain.

12. - Trap and defects are same thing. Right? Please maintain the consistency. 

13. Energetic Particles will induce defects thoroughout the material/device. How the particular positions viz. A,B,C,D,E in the device are selected? Explain. 

14. How defects cause increment in the breakdown voltage? Please explain. 

Author Response

Thank you for your valuable comment.

We revised with you comment. Please find out the attachment.

Reviewer 2 Report

Major Comments

  1. Authors should have included a section correlating the radiation dose with trap concentration, type of trad defects and energy level.
  2. Figure 2(b) - The results should be extended to Vgs=25 Volts in order to substantiate the inference made by the authors that  "In positions D and E, on-current starts to drop for Vg > 10 V and Vg 87 > 12 V, respectively. As the location is relatively far from the gate oxide interface, the charge trapping become effective at higher gate voltage".
  3. As per Figure 6(b) - "..the deeper the trap energy level and the larger the trap concentration at position E, the higher the increase in breakdown voltage..". This means that Position E is important and should a much more detailed study for Position C-E is required 
  4. A separate section must be included for explaining the impact of irradiation on the
  1. static (dc) [ i.e. gate leakage current,  gate-source threshold voltage, drain leakage current,  drain-source on-state voltage and source-drain diode voltage results],
  2. pulse characteristics of the device
  3. RF characteristics of the device
  4. High-temperature reverse-bias (HTRB) & High-temperature gate-bias (HTGB) simulation studies.

Author Response

(The authors gave the same response as above.)

Round 2

Reviewer 1 Report

Authors have made sufficient responses to the quesries raised by reviewer. Based on those, the work may be accepted.